# The NAC transcription factors SNAP1/2/3/4 are central regulators mediating high nitrogen responses in mature nodules of soybean

Xin Wang [1,7], Zhimin Qiu [2,3,7], Wenjun Zhu [2,7], Nan Wang [4,7], Mengyan Bai [2,3], Huaqin Kuang[1], Chenlin Cai [5], Xiangbin Zhong[5], Fanjiang Kong [1], Peitao Lü [3,6] ✉ & Yuefeng Guan [1,3] ✉

Legumes can utilize atmospheric nitrogen via symbiotic nitrogen fixation, but this process is inhibited by high soil inorganic nitrogen. So far, how high nitrogen inhibits $N_2$ fixation in mature nodules is still poorly understood. Here we construct a co-expression network in soybean nodule and find that a dynamic and reversible transcriptional network underlies the high N inhibition of $N_2$ fixation. Intriguingly, several NAC transcription factors (TFs), designated as Soybean Nitrogen Associated NAPs (SNAPs), are amongst the most connected hub TFs. The nodules of *snap1/2/3/4* quadruple mutants show less sensitivity to the high nitrogen inhibition of nitrogenase activity and acceleration of senescence. Integrative analysis shows that these SNAP TFs largely influence the high nitrogen transcriptional response through direct regulation of a subnetwork of senescence-associated genes and transcriptional regulators. We propose that the SNAP-mediated transcriptional network may trigger nodule senescence in response to high nitrogen.

Soybean (*Glycine max* L. Merr.) can fix atmospheric nitrogen (N) gas inside root nodules through symbiotic interactions with rhizobia, a process called symbiotic nitrogen fixation (SNF). SNF helps fulfill the high nitrogen demand of plants and improves soil fertility, making soybean a cornerstone crop in sustainable agricultural practice. However, when soil is heavily fertilized with inorganic N, SNF is inhibited[1–4]. This phenomenon avoids the unnecessary consumption of carbon (C) in nodules but limits the contribution of SNF to soybean production in agricultural systems[5]. Thus, breeding soybean varieties with $N_2$ fixation ability that is stable across a range of N levels may provide environmental benefits through decreasing N-inputs.

The mechanisms underlying high N inhibition of nodulation and SNF in legume plants have been pursued for over a century. High N, regardless of the chemical form, can inhibit multiple aspects of SNF, including repression of nodule number and organogenesis, inhibition of nitrogenase activity, and acceleration of nodule senescence[6–9]. The molecular mechanism for N-control of nodule number has been relatively well characterized. The autoregulation of nodulation (AON) is a conserved long-distance signaling cascade negatively controlling nodulation in legumes. which requires the upregulation of CLAVATA3/EMBRYO SURROUNDING REGION-RELATED (CLE) peptides that are perceived by a leucine-rich repeat receptor-like kinase (LRR-RLK)[4,10].

[1]Guangdong Provincial Key Laboratory of Plant Adaptation and Molecular Design, Innovative Center of Molecular Genetics and Evolution, School of Life Sciences, Guangzhou University, Guangzhou 510006, China. [2]College of Life Sciences, Fujian Agriculture and Forestry University, Fuzhou, Fujian 350002, China. [3]FAFU-UCR Joint Center for Horticultural Biology and Metabolomics, Fujian Agriculture and Forestry University, Fuzhou 350002, China. [4]School of Life Sciences, Inner Mongolia University, Hohhot 010000, China. [5]College of Resources and Environment, Fujian Agriculture and Forestry University, Fuzhou, Fujian 350002, China. [6]College of Horticulture, Fujian Agriculture and Forestry University, Fuzhou, Fujian 350002, China. [7]These authors contributed equally: Xin Wang, Zhimin Qiu, Wenjun Zhu, Nan Wang. ✉e-mail: ptlv@fafu.edu.cn; guan@gzhu.edu.cn

When soil N availability is high, nitrate-responsive CLE peptides are induced (GmNIC1a/1b in soybean, PvNIC1 in common bean, LjCLE-RS2/RS3/40 in *Lotus japonicus*, and MtCLE34/35 in *Medicago truncatula*) to inhibit nodule numbers[11–13]. NIN-LIKE PROTEIN (NLP) transcription factors (TFs), represented by LjNLP4 in *Lotus japonicus* and MtNLP1 in *Medicago truncatula*, localize within nuclei in response to high nitrate and negatively regulate nodulation by inducing negative regulators and repressing positive regulators of nodulation[14–16]. For instance, cytokinin biosynthesis is inhibited by NLP-mediated signaling in N-repressed organogenesis in *Lotus japonicus*[16].

The suppression of $N_2$ fixation activity in mature nodule is another critical aspect of the inhibitory effects of high soil N to SNF. In soybean, $N_2$ fixation activity can be inhibited within 48 h of nitrate application, and the inhibitory effect is reversible[17]. Such an inhibitory effect is accompanied by repression of C allocation to the nodule and increased oxygen diffusion resistance[8]. Following the inhibition of $N_2$ fixation activity, nodule senescence is accelerated by high N. The most visible sign of senescence is a change of the nodule color from red (which signifies functional $O_2$-bound leghemoglobin) to green (resulting from leghemoglobin nitration and breakdown)[18,19]. In addition, the level of reactive oxygen and nitrogen species increases, and symbiosomes and bacteroids are degraded[18]. These physiological and developmental responses of nodules to high N appear similar to the natural senescence process yet occur much quicker[6,18].

The inhibition of nodule functions and promotion of nodule senescence by N represent distinct phenomena, as nitrate generally triggers increased plant growth, root foraging, and delayed leaf senescence[20]. This indicates that legume nodules may be regulated by a signaling network that is distinctive from the previously characterized N-response gene regulatory networks (e.g. in *Arabidopsis thaliana*)[20]. Indeed, in mature nodules of *Medicago truncatula*, high nitrate tends to down-regulate genes related to maintenance of nodule cell function, and up-regulate senescence related genes[21]. However, the regulatory mechanisms linking N status and nodule function are still poorly understood.

In this study, we sought to define the specific molecular mechanisms correlated with SNF repression by N. We constructed a transcriptional regulatory network and identified a set of NAC TFs as highly connected hubs. Multiple lines of evidence showed that these TFs play key roles in regulating nodule responses to high N.

## Results

### N suppression of nodule function is reversible in soybean

To understand N responses of mature soybean nodules, we employed a hydroponic culture system for reversible N treatment. Soybean plants were grown for 28 days under low N (0.5 mM total N, including 0.4 mM nitrate + 0.1 mM ammonium) to promote nodule development and then transferred to high N medium (10 mM total N, 8 mM nitrate + 2 mM ammonium). Nodule samples were collected at 1, 3, or 5 days after the high N treatment (1HN, 3HN or 5HN) (Fig. 1a). A subset of the plants were returned to low N medium 3 days after treatment and samples were collected 2 days later (3HN2LN) (Fig. 1a). Control groups in low N were sampled and compared to treatment groups at equivalent time points (Fig. 1a). We performed an acetylene reduction assay (ARA) to estimate potential nitrogenase activity in response to N treatment. ARA activities were not significantly affected in 1HN, but decreased by ~50% in 3HN and 5HN nodules (Fig. 1b). A return to low N resulted in a full recovery of ARA activity in 3HN2LN nodules (Fig. 1b). These results are consistent with previous findings that high N inhibition of nodule function relies on a dynamic and reversible mechanism[22].

We also dissected the N responsive physiological changes in the nodule. The red color of nodule was similar between low N control and 1HN nodules, yet was decreased in 3HN and 5HN nodules (Fig. 1c and Supplementary Fig. 1a). In 3HN2LN nodules, the red color was partially recovered in comparison with 3HN and 5HN nodules (Fig. 1c and Supplementary Fig. 1a). This visual observation was consistent with the measurement of red leghemoglobin content (Fig. 1d and Supplementary Fig. 2a). Rhizobia viability, indicated by SYTO9 staining[23], showed similar trend with red leghemoglobin content (Fig. 1e, f, Supplementary Fig. 2b and Supplementary Fig. 3a).

For detailed cell morphology observation, we next performed transmission electron microscope (TEM). The infected cells of nodules at low N were packed with symbiosomes containing bacteroids (Fig. 1g and Supplementary Fig. 4a). In contrast, in 3HN nodules, large lytic vacuolar compartments were observed as a result of symbiosomes fusion[24]. Moreover, the formation of symbiosomes were notably different, and polymer poly-β-hydroxybutyrate (PHB) was over accumulated in the bacteroids (Fig. 1g, Supplementary Fig. 4a and Supplementary Fig. 5a). These symptoms, reported as features of senescent nodules[25,26], were more severe in 5HN nodules (Fig. 1g, Supplementary Fig. 4a and Supplementary Fig. 5a). In 3HN2LN nodules, lytic vacuoles still existed, yet the symbiosome formation was partially recovered (Fig. 1g, Supplementary Fig. 4a and Supplementary Fig. 5a). Taken together, in response to reversible high N treatment, mature nodules exhibited senescent symptoms along with changes of nitrogenase activity in our hydroponic system.

### Construction of a nodule co-expression network for high N responses

To understand the molecular basis behind the N response of mature nodules, we performed RNA-seq of nodules samples under the different treatments described above (Supplementary Fig. 6). Differentially expressed genes (DEGs) were identified for each time point. Overall, massive transcriptional changes were observed in response to N treatments in nodules, with a total of 13,446 DEGs (28.6% of expressed genes) being affected by the various N regimes (Supplementary Data 1). In comparison with LN controls, just 170 DEGs were observed in 1HN nodules, while 7749 and 9422 DEGs were found at 3HN and 5HN, respectively. Removal of nitrogen (3HN2LN) dramatically decreased the number of DEGs (1423) compared to 3HN (Fig. 2a, Supplementary Fig. 7 and Supplementary Fig. 8). The observed changes in gene expression parallel the changes seen in ARA activity, suggesting that dynamic transcriptional changes underly the reversible suppression of nodule activity by high N.

To explore the genes associated with the modulation of SNF by N status, we constructed a weighted gene co-expression network using the full transcriptome dataset and integrating the ARA activity. We generated eight expression modules and calculated the correlation between each module and ARA activity (Fig. 2b, c and Supplementary Data 2). Among these, the 'turquoise' module, which contains 5330 genes with N-repressed expression, is the largest module and most positively correlated with ARA activity (Fig. 2b, c and Supplementary Data 2). The 'blue' module, with 2349 genes, is the most negative correlated with ARA activity, and the genes in this module are induced by high N exposure (Fig. 2b, c and Supplementary Data 2).

We performed Gene Ontology (GO) enrichment analysis, and found contrasting regulation of biological processes in the two modules. Enrichment of GO terms in the N-inhibited 'turquoise' module include "oxidation reduction", "microtubule-based movement", "spindle organization", "DNA replication initiation", "glucose metabolic process" and "cellular amino acid biosynthetic process" (Fig. 2e and Supplementary Data 3). In the N-induced 'blue' module, enrichment was seen for genes related to "regulation of transcription", "trehalose biosynthetic process" and "autophagic vacuole assembly" (Fig. 2e and Supplementary Data 4). Early studies showed that trehalose abundance and autophagic activity increase during nodule senescence[27,28]. Thus, the upregulation of these processes may be related to accelerated nodule senescence. The enrichment of "regulation of transcription" particularly drew our attention, as this

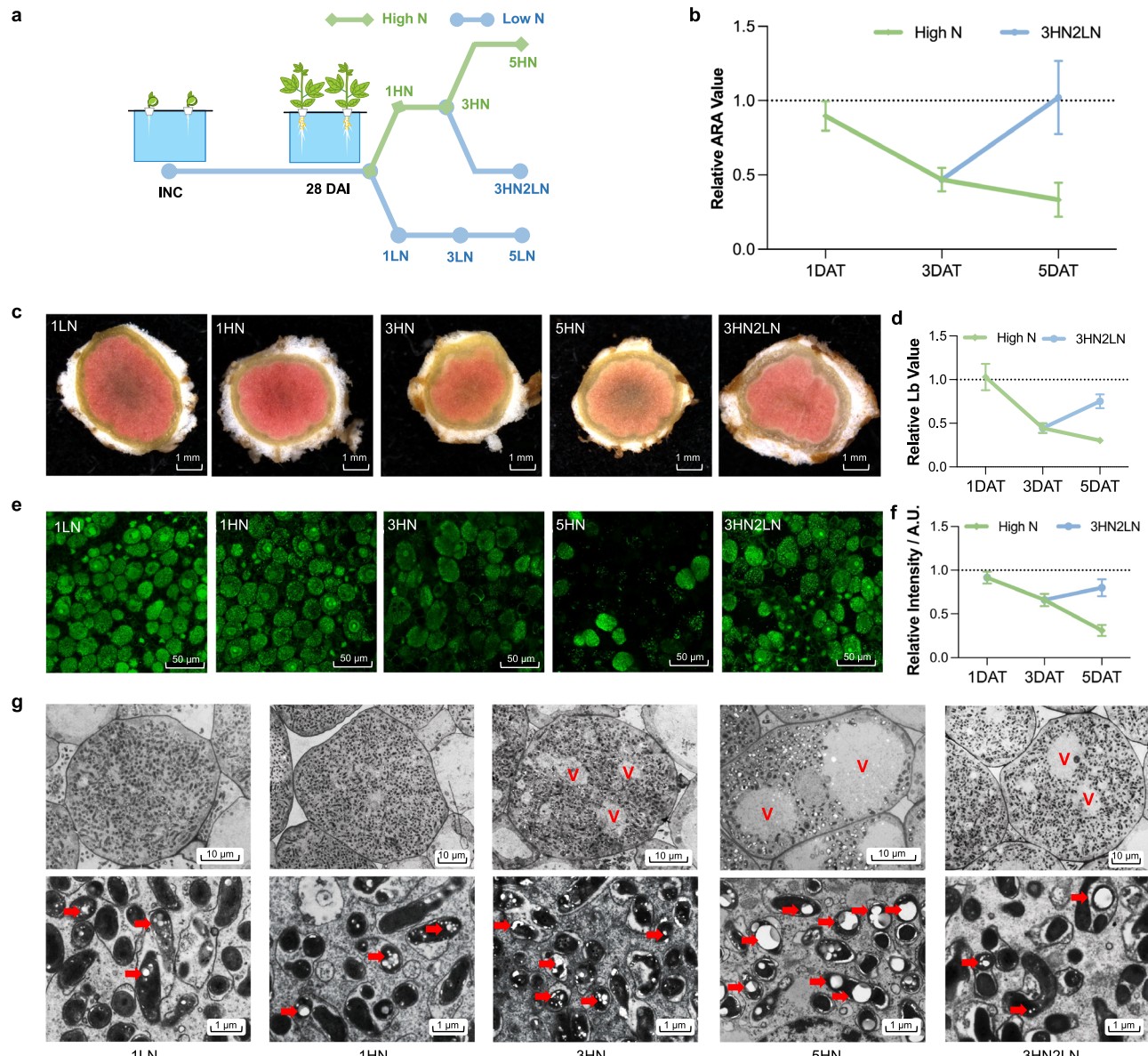

**Fig. 1 | Physiological responses of mature soybean nodules to different N treatments in hydroponic culture. a** A hydroponic culture system for reversible N treatment. INC inoculation, DAI day after inoculation, HN high N treatment, LN low N treatment; numbers indicate the days after treatment. **b** ARA activity of mature nodules under different N treatments. ARA acetylene reduction assay, DAT day after treatment. Data are represented as mean ± SD (*n* = 8). **c** Cross section observations of mature nodules under different N treatments. Scale bar = 1 mm. More sections were shown in Supplementary Fig. 1a. **d** Relative leghemoglobin change of nodule samples under different N treatments. Leghemoglobin content of nodule samples under different N treatments were shown in Supplementary Fig. 2a. Data are represented as mean ± SD (*n* = 12). **e** SYTO9 staining for rhizobia in mature

nodules under different N treatments. More sections of SYTO9 staining were shown in Supplementary Fig. 3a. Scale bar = 50 μm. **f** Relative fluorescence intensity of SYTO9 staining for rhizobia in mature nodules under different N treatments. Fluorescence intensity of SYTO9 staining for rhizobia in mature nodules under different N treatments were shown in Supplementary Fig. 2b. Data are represented as mean ± SD (*n* = 20). **g** Transmission electron micrographs of mature nodules under different N treatments. More micrographs were shown in Supplementary Fig. 5b. Up panel: Scale bar = 10 μm; down panel: Scale bar = 1 μm. V lytic vacuolar compartments. PHB was indicated by red arrow. Source data underlying **b**, **d**, and **f** are provided as a Source Data file.

suggested the crucial roles of TFs in the N-induced transcriptome reprogramming. To identify influential TFs in this module, we calculated intramodular connectivity (sum of the weight of intramodular edges) and gene trait significance (correlation between gene expression and ARA activity) for ranking TFs in the 'blue' module (Supplementary Data 5). The TFs with the highest intramodular connectivity and gene trait significance were selected as hubs. These TFs showed similar N responses, each being significantly upregulated in 3HN and 5HN nodules with attenuated expression upon N-removal (3HN2LN) (Supplementary Data 5).

**N-inhibition of nodule activity and N-induced senescence were alleviated in *snap1/2/3/4* quadruple mutants**

Interestingly, six of the top eight most connected TFs are NAC domain-containing proteins (Glyma.19G108800, Glyma.01G051300, Glyma.02G109800, Glyma.12G221400, Glyma.16G043200 and Glyma.13G280000), which all belong to the SNAC-B/NAP sub-family[29]. In total, 11 members of this subfamily were identified in the 'blue' module (Supplementary Data 5), all of which showed induction by high N that could be reduced by N removal (Fig. 3a). This result suggested that members of the SNAC-B/NAP sub-family potentially function in

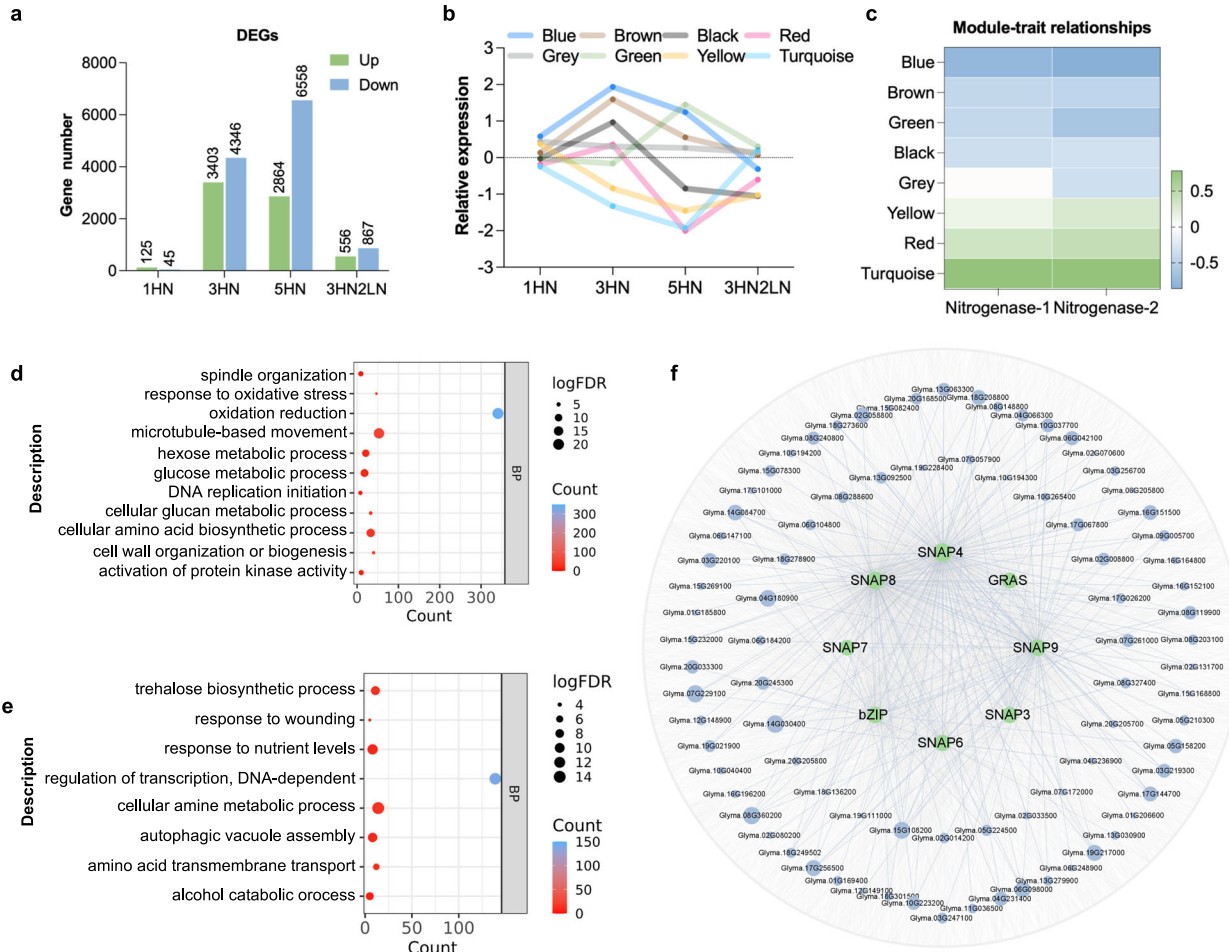

**Fig. 2 | A co-expression network of mature soybean nodules in response to high N treatment. a** The number of DEGs under various N treatments. See also Supplementary Fig. 7 and Supplementary Fig. 8. **b** The expression patterns of co-expressed modules. Colors represent different modules. Data are represented as mean log2(fold change) of genes in same module. **c** Correlations between modules and ARA activity. Two biological replicates were shown. **d, e** GO enrichment analysis of genes in (**d**) turquoise and (**e**) blue modules. BP: biological process. All significantly enriched items are listed in Supplementary Data 3 and 4. **f** Subnetwork of top eight hub TFs and their neighbors in the blue module. Hub TFs and their neighbors with a weighted correlation >0.27 were selected for ease of visualization. Green circles indicate the hub TFs and blue circles indicate individual genes. Source data underlying **b** is provided as a Source Data file.

N-suppression of nodule function. We designated these N induced NAC genes as *SNAP1-11* (Soybean Nitrogen Associated NAP subfamily genes).

Among the *SNAP* genes, *SNAP3* exhibited relatively higher expression in the nodules (Supplementary Fig. 9). In situ hybridization showed that its transcript accumulates in the infected cells of mature nodule under high N (Supplementary Fig. 10). Considering the high homology among *SNAP* members (Supplementary Fig. 11), we performed multiplex mutagenesis of *SNAP1/2/3/4* to determine their biological function. The conserved N-terminal DNA-binding domains of *SNAP1/2* and *SNAP3/4* each targeted by three sgRNAs (Fig. 3b and Supplementary Table 1). The two multiplex vectors were pooled and transformed into soybean Williams 82 by *Agrobacterium tumefaciens*. A total of 30 transgenic lines were produced, and from these two independent T2 *snap1/2/3/4* quadruple mutant lines with frameshift mutations or large fragmental deletions at all target genes were identified, designated as *snap1/2/3/4-1* and *snap1/2/3/4-2* (Fig. 3c). RT-qPCR analysis confirmed that the transcript levels of all target genes were greatly reduced in nodules of these mutants (Fig. 3d).

We next examined the nodulation potential of the *snap1/2/3/4-1* and *snap1/2/3/4-2* mutants. At low N conditions, vegetative growth of the *snap1/2/3/4* mutants was apparently similar to the wild type (WT) (Supplementary Fig. 12), and the nodule number and average nodule weight was not statistically different from the WT (Fig. 3e, f).

Nitrogenase activity was also indistinguishable between the WT and mutants at low N (Fig. 3g). Nevertheless, after 5HN exposure, the ARA activities of *snap1/2/3/4-1* and *snap1/2/3/4-2* nodules were higher than the WT (Fig. 3g). Moreover, the content of red leghemoglobin in *snap1/2/3/4* mutant nodules was similar to WT at low N, yet was higher than WT at 5HN (Fig. 4a, b and Supplementary Fig. 1b). The mutant nodules also displayed less sensitivity to high N-induced rhizobia death as indicated by SYTO9 staining (Fig. 4c, d and Supplementary Fig. 3b). The TEM observation also showed that, in 5HN *snap1/2/3/4* nodules, the symbiosome formation alteration and large lytic vacuolar compartments were less obvious than in 5HN WT nodules. (Fig. 4e, Supplementary Fig. 4b and Supplementary Fig. 5b). These results indicate that *snap1/2/3/4* mutant nodules are normal at low N conditions but are less sensitive to high N inhibition of nitrogenase activity and acceleration of nodule senescence.

**Combined RNA-seq and ChIP-seq to identify direct and indirect downstream genes of SNAP1/2/3/4 TFs in mature nodules**

To uncover the direct targets of SNAP1/2/3/4 TFs in N-treated nodules, we performed RNA-seq and ChIP-seq analyses. RNA-seq analysis was performed with WT and *snap1/2/3/4-1* nodules at 5LN and 5HN conditions (Supplementary Figs. 13–15). A total of 10,273 DEGs were identified between 5LN and 5HN WT nodules (Fig. 5a and Supplementary Data 6), similar to the first RNA-seq experiment (Fig. 2a). However,

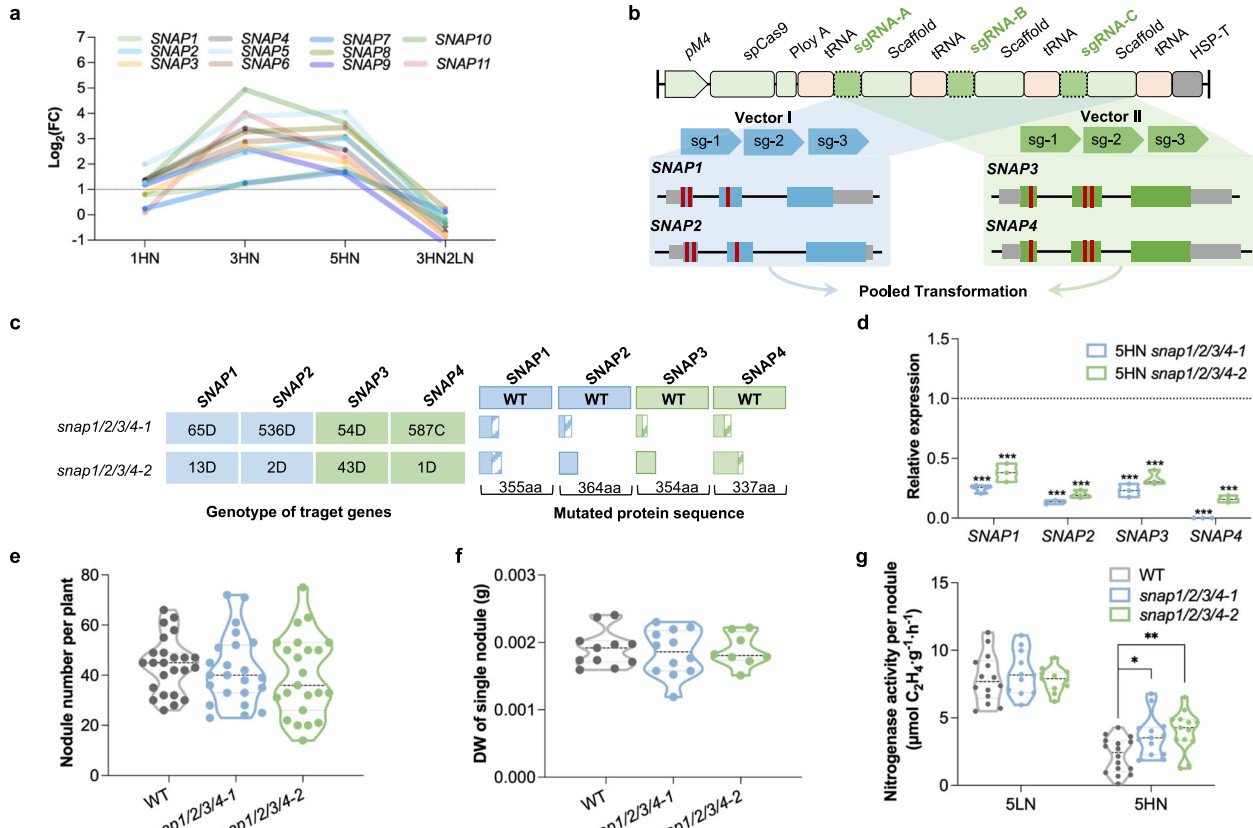

**Fig. 3 | *snap1/2/3/4* quadruple mutations alleviate N-inhibition of nodule activity. a** Relative expression of SNAPs across N treatments. FC fold change. **b** sgRNA designed for SNAP1/2/3/4 genome editing. The conserved N-terminal DNA-binding domain of SNAP1/2 and SNAP3/4 were each targeted by three sgRNAs. **c** Mutations in SNAP genes in two independent quadruple mutant lines (*snap1/2/3/4-1* and *snap1/2/3/4-2*). Numbers indicate the number of base pairs or aa (amino acid). D: deletion; C: complex mutant type, including 585 bp inversion, 1 bp deletion and 1 bp insertion. **d** Relative expression of SNAP1/2/3/4 in two *snap1/2/3/4* mutants. Three biological replicates were shown. **e** Nodule number of WT and two *snap1/2/3/4* mutants under LN condition. Each point represents a single plant. **f** Nodule dry weight of WT and two *snap1/2/3/4* mutants under LN condition. Each point represents a single plant. **g** ARA activity of WT and two *snap1/2/3/4* mutants under 5HN and 5LN. Each point represents a single nodule. Statistically significant differences performed in this figure used Student's *t*-test (two-tailed). *$P < 0.05$, **$P < 0.01$. Source data underlying **a** and **d**–**g** are provided as a Source Data file.

between 5LN and 5HN *snap1/2/3/4* nodules, the number of DEGs was only 6689 (Fig. 5a and Supplementary Data 7). For the shared 5HN vs 5LN DEGs, the average log2(fold change) was significantly reduced from 3.00 in WT to 2.61 in *snap1/2/3/4* mutant nodules (Fig. 5b). These results indicate that the transcriptional responses to high N are reduced in *snap1/2/3/4* nodules.

We next compared the transcriptome between *snap1/2/3/4* mutant and WT nodules at low N and high N conditions, respectively. Compared to 1096 DEGs at low N condition, we found more DEGs (3282) between *snap1/2/3/4* and WT in 5HN nodules (Fig. 5c, Supplementary Data 8 and 9). This result indicated that SNAP1/2/3/4 makes more impact on nodule transcriptome after high N exposure. Notably, over 69% of "*snap1/2/3/4* VS WT" DEGs at 5HN (2,279/3,282) were identified as N-responsive genes in WT, and 41% of DEGs (1,361/3,282) were found in either the 'blue' ($n = 697$) or 'turquoise' ($n = 664$) modules. Interestingly, for the genes upregulated in *snap1/2/3/4* vs WT in the 5HN treatment, enriched GO terms included "cellular amino acid biosynthetic process", "glucose metabolic process", "oxidation reduction" and "protein amino acid phosphorylation" (Fig. 5d and Supplementary Data 10). Most of these terms were also enriched in the N-repressed expression module blue in WT (Fig. 2d). Taken together, these results suggest that SNAP1/2/3/4 TFs mainly influence the N-responsive transcriptome in mature nodules.

To identify the genome-wide binding sites of SNAP1/2/3/4 transcription factors, we performed ChIP-seq assays with two

biological replicates of transgenic hairy roots expressing FLAG-tagged SNAP1/2/3/4 TFs. We identified 3083, 2814, 1692, and 1909 putative target genes for SNAP1/2/3/4 TFs, respectively (Supplementary Fig. 16 and Supplementary Data 11). The SNAP1/2/3/4 binding peaks were highly enriched ~500 bp upstream of the transcription start sites (TSS) of genes (Fig. 5e). We also found that the previously reported NAC core binding motifs with the consensus CACGT, are abundant in SNAP1/2/3/4 binding sequences (Fig. 5f and Supplementary Data 12). By searching for overlapping targets among the four SNAP TFs, we identified 771 genes targeted by all four SNAP TFs, accounting for 25.0% of SNAP1, 27.4% of SNAP2, 45.6% of SNAP3, and 40.4% of SNAP4 targets (Fig. 5g). In total of 4689 genes were bound by at least one of the SNAP1/2/3/4 TFs, and 2547 genes were targeted by at least two SNAP TFs. This result suggests that SNAP1-4 TFs play mostly redundant roles in transcriptional regulation, but does not exclude specific roles.

As our data suggest that SNAP TFs have primary roles in high N responses, we integrated our RNA-seq for N-responsive DEGs between *snap1/2/3/4* and WT 5HN nodules with the ChIP-seq data. This led us to identify 388 genes directly regulated by SNAP1/2/3/4 that were N-responsive (223 activated and 165 repressed). The remaining 1891 N-responsive DEGs that were not identified as direct targets (884 repressed and 1007 activated) were defined as indirectly regulated (Fig. 5h). Therefore, SNAPs influence 2279 N-responsive genes, 388 of which are directly regulated.

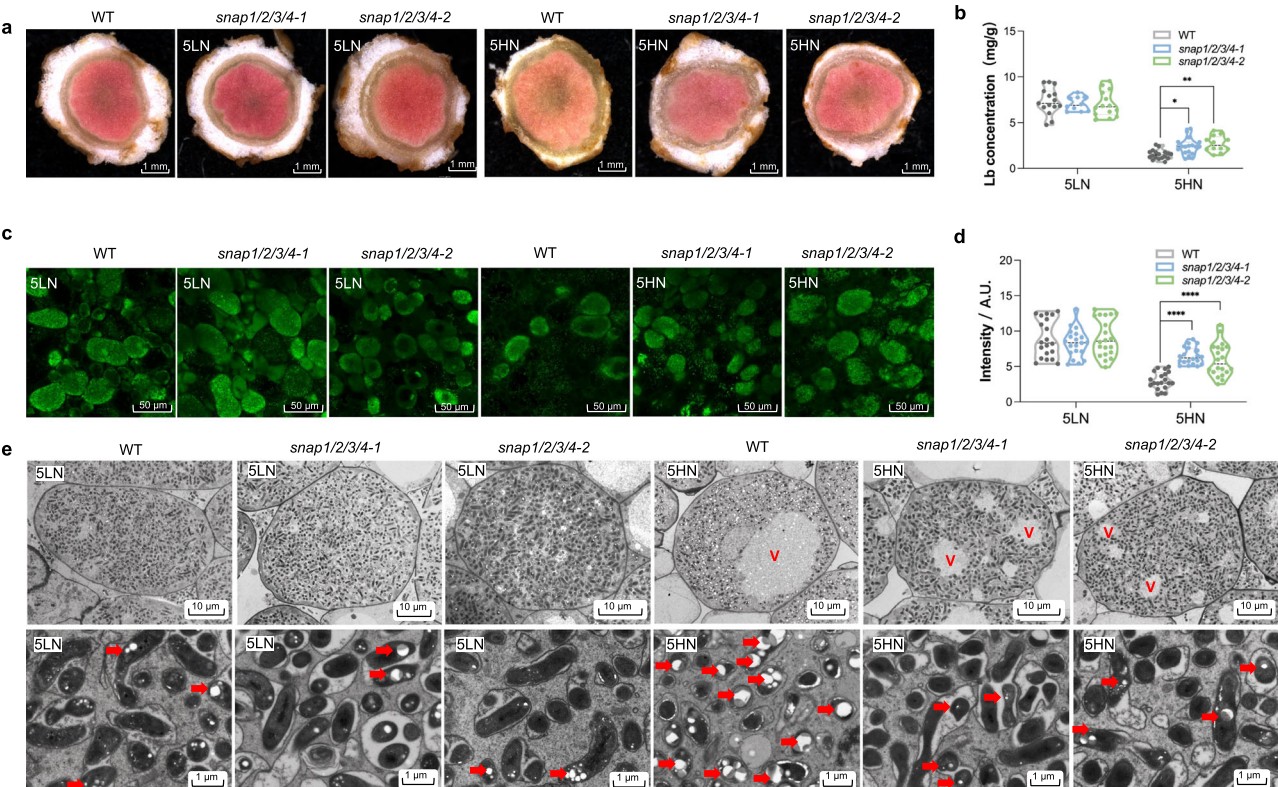

**Fig. 4 | *snap1/2/3/4* quadruple mutations alleviate N-induced nodule senescence. a** Cross section observation of mature nodules from WT and two *snap1/2/3/4* mutants under 5LN and 5HN treatments. More sections were shown in Supplementary Fig. 1b. **b** Statistical summary of leghemoglobin content of nodule samples under 5LN and 5HN treatments. At least 10 values of each sample were shown. **c** SYTO9 staining for rhizobia in mature nodules from WT and two *snap1/2/3/4* mutants under 5LN and 5HN treatments. More sections of SYTO9 staining were shown in Supplementary Fig. 3b. **d** fluorescence intensity of nodule samples under 5LN and 5HN treatments. The 20 values of each sample were shown. **e** Transmission electron micrographs of mature nodules of WT and two *snap1/2/3/4* mutants under 5LN and 5HN treatments. More micrographs were shown in Supplementary Fig. 5b. Up panel: Scale bar = 10 μm; down panel: Scale bar = 1 μm. V lytic vacuolar compartments. PHB was indicated by red arrow. Statistically significant differences performed in this figure used Student's *t*-test (two-tailed). *$P < 0.05$, **$P < 0.01$, ***$P < 0.001$, ****$P < 0.0001$. Source data underlying **b** and **d** are provided as a Source Data file.

## SNAP1/2/3/4 mediate a transcriptional cascade in N-triggered senescence of mature nodules

To estimate the overall contribution of SNAP1/2/3/4 TFs on nodule N-responsive genes, we built a regulatory subnetwork using N-responsive genes that were directly bound and regulated by SNAP TFs (Fig. 6a). By comparing our ChIP-Seq data with publicly available transcriptome data for senescent soybean nodules (senescence-associated genes; SAGs), we observed they largely overlapped (166/223, 74.4%) with the SNAP directly activated targets in nodules (Supplementary Data 13)[30]. GO term analysis showed that genes in this subnetwork are significantly enriched in terms with "transcription regulation", including 33 TFs (26 activated and 7 repressed), 25 of which are SAGs in nodules (Supplementary Fig. 17 and Supplementary Data 14). Particularly, SNAPs directly activate 7 NAC, 7 ERF, and 4 WRKY family TFs (Fig. 6a, b, Supplementary Data 17 and Supplementary Data 14). These TF genes are induced in WT 5HN but to a lesser extent in *snap1/2/3/4* 5HN nodules (Fig. 6c), and are mostly SAGs in senescent nodules (7 NACs, 7 ERFs, and 3 WRKYs)[30]. NAC, ERF, and WRKY family TFs are enriched for senescence-promoting TFs in many plant species. This is consistent with the common function of NAC TFs in promoting plant senescence and the alleviation of N induced nodule senescence in *snap1/2/3/4* mutants.

Ethylene was shown to play positive roles in nodule senescence in *Medicago truncatula*, and ERFs are key components of ethylene signaling[18]. Here we identified that 7 ERFs were directly activated by one or more SNAP TFs (Fig. 6a, b, Supplementary Fig. 18 and Supplementary Data 11). WRKY6, WRKY22 and WRKY33 in Arabidopsis were shown to promote leaf senescence[31–33]. We found that these WRKY homologs in soybean, which are GmWRKY6 (Glyma.15G110300 and Glyma.09G005700), GmWRKY22 (Glyma.16G031900), and GmWRKY33 (Glyma.09G280200), were targeted by SNAP1/2, SNAP1/2/4 and SNAP1/2/3/4 respectively. (Fig. 6a, b, Supplementary Fig. 17 and Supplementary Data 11). Interestingly, we found that SNAP TFs cross activate each other. For example, SNAP1/2/3 all target *SNAP8*, SNAP1 targets *SNAP2* and *SNAP10*, and SNAP3 targets *SNAP1* (Fig. 6a, b and Supplementary Data 11). This is further supported by the finding that in *snap1/2/3/4* mutants, *SNAP5-11* are also down-regulated in 5HN nodules (Supplementary Fig. 19).

There were several types of SAG genes specifically regulated by SNAPs. Specifically, there were three protein phosphatase 2 C family proteins (PP2Cs), Glyma.06G238200, Glyma.17G038000, and Glyma.20G131500, activated by SNAP1/2/3/4, SNAP2/4 and SNAP1 (Fig. 6a, b and Supplementary Data 11), respectively. There were also two gamma vacuolar processing enzymes (VPEs), Glyma.14G092800 and Glyma.17G230700, activated by SNAP3 and SNAP1 (Fig. 6a, b and Supplementary Data 11), respectively. Another group included five Chaperone DnaJ-domain superfamily proteins (HSP40s), four of which (Glyma.01G166000, Glyma.07G021800, Glyma.11G077400 and Glyma.16G127700) are targeted by four SNAP TFs and Glyma.11G105800 which is targeted by SNAP1/2 (Fig. 6a, b and Supplementary Data 11). The homologs of these target genes were shown to regulate senescence in Arabidopsis leaves or in *Medicago truncatula* nodules[34–36]. Based on the above evidence, we hypothesize that SNAP TFs may trigger a transcriptional cascade to initiate nodule senescence under high N (Fig. 7).

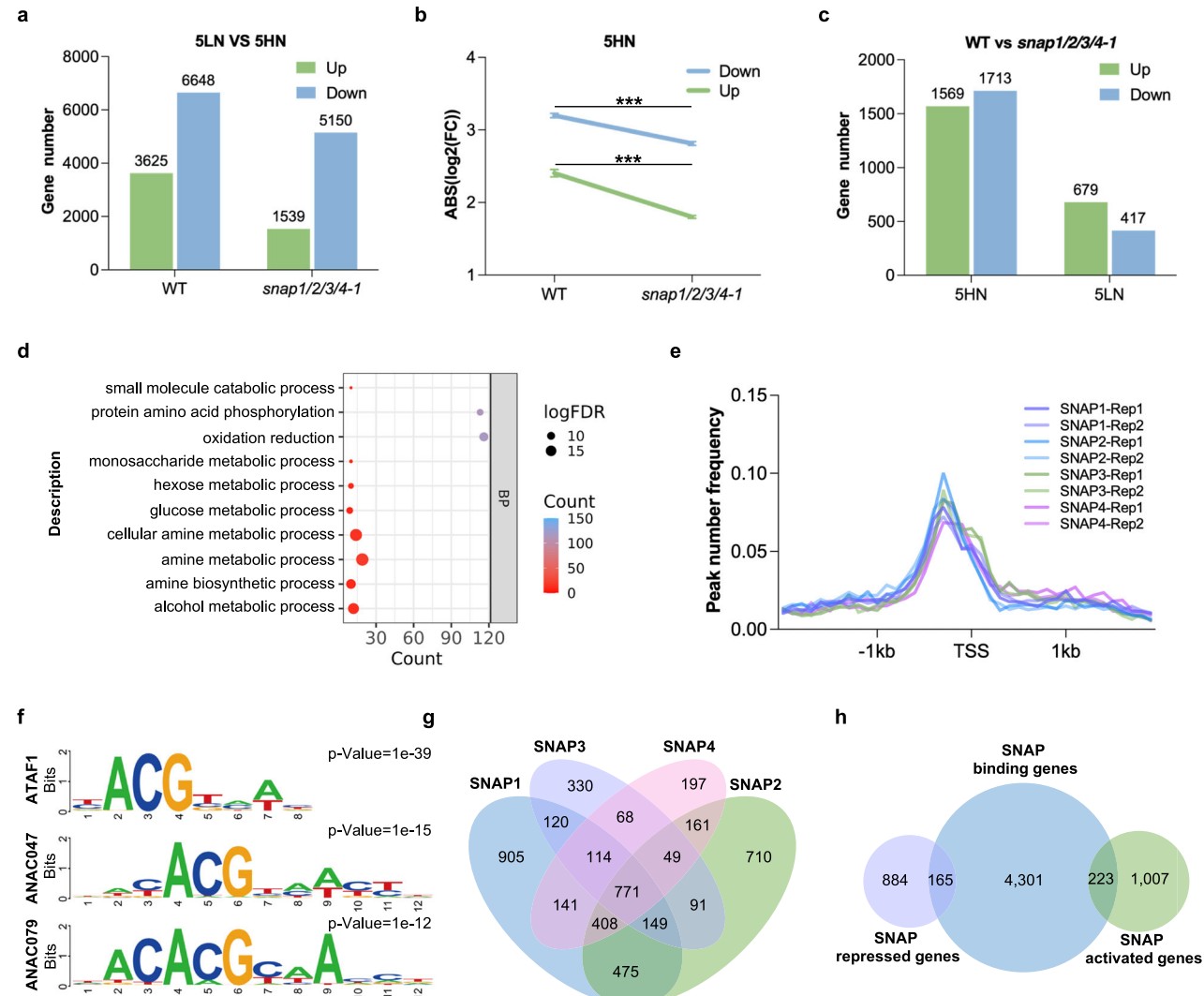

**Fig. 5 | RNA-seq and ChIP-seq analysis reveal genes downstream of SNAP TFs. a** Number of DEGs between 5LN and 5HN in nodule samples of WT and the *snap1/2/3/4* mutant. See also Supplementary Fig. 14 and Supplementary Fig. 15. **b** Average log2(FC) of 5HN vs 5LN DEGs shared by WT and *snap1/2/3/4*. FC: fold change; ABS: absolute. Data are represented as mean log2(FC) ± SEM of genes. Statistically significant differences performed in this figure used Student's *t*-test (two-tailed). ****P* < 0.001. **c** Number of DEGs between WT and *snap1/2/3/4* mutant under LN and HN conditions. See also Supplementary Fig. 14 and Supplementary Fig. 15. **d** GO analysis of upregulated genes between *snap1/2/3/4* mutant vs WT under 5HN condition. BP: biological process. All significantly enriched items are listed in Supplementary Data 10. **e** SNAP1/2/3/4 binding sites were highly enriched ~500 bp upstream of the TSS. **f** Three Arabidopsis NAC-like motifs were detected by HOMER in 200-bp flanking sequences around the SNAP1 binding peaks. The predicted motifs of all SNAP TFs are listed in Supplementary Data 12. **g** Venn diagram of SNAP1/2/3/4 targets. **h** Venn diagram of SNAP targets and N-responsive genes. Source data underlying **b** and **e** are provided as a Source Data file.

## Discussion

Here we present multiple lines of evidence that SNAP1/2/3/4 TFs act as key hubs in the gene network controlling N response in mature nodules. These NACs were among the top connected TFs in the soybean nodule co-expression networks across N treatments. We found that SNAP1/2/3/4 mediate the N inhibition of nitrogenase activity and acceleration of nodule senescence and are largely required for the reprogramming of the nodule N-response transcriptome. Moreover, these SNAPs were found to directly regulate a subnetwork enriched in senescence related genes including TFs in the NAC, WRKY and ERF families. Therefore, SNAP1/2/3/4 likely act as master regulators in a transcriptional cascade that defines the gene network in mature nodules, which conditions the inhibition of nodule function and acceleration of senescence in response to high N.

How plants respond to fluctuating N status is an important biological question and has been extensively elucidated[20,37]. In Arabidopsis, the nitrate sensor NLP7 acts as part of a $Ca^{2+}$-CPK-NLP signaling cascade that is central to the nitrate response[38,39]. The

downstream secondary TFs, such as TGA1, bZIP1, TCP20, HRS1 and ABF2 were identified to directly bind in vivo to the promoters of cognate nitrate-responsive target genes[20,37,39–42]. In legumes, NLPs are involved in the nitrate-induced control of multiple processes involved in root nodule symbiosis, including nitrogen fixation[14,15,43]. In this work, we found that the high N response of mature nodules may be mediated by a SNAPs-centered transcriptional network that activates WRKY, ERF and NAC family TFs. It is interesting that SNAP1/2/3/4 directly binds to a small portion of SNAPs influenced N-responsive genes. Therefore, SNAP1/2/3/4 may largely influence N-responsive genes by orchestrating the TF hierarchy in the nodules. Alternatively, this is possibility of transient bindings which may affect target gene expression yet elude biochemical detection.

For the SNAP1/2/3/4 binding targets without altered expression in *snap1/2/3/4/* nodules, these genes maybe be regulated by SNAPs in other tissues or conditions. Otherwise, the expression of these target genes may be not sufficiently affected by snap mutations, due to functional redundancy or compensation effect. More studies on this

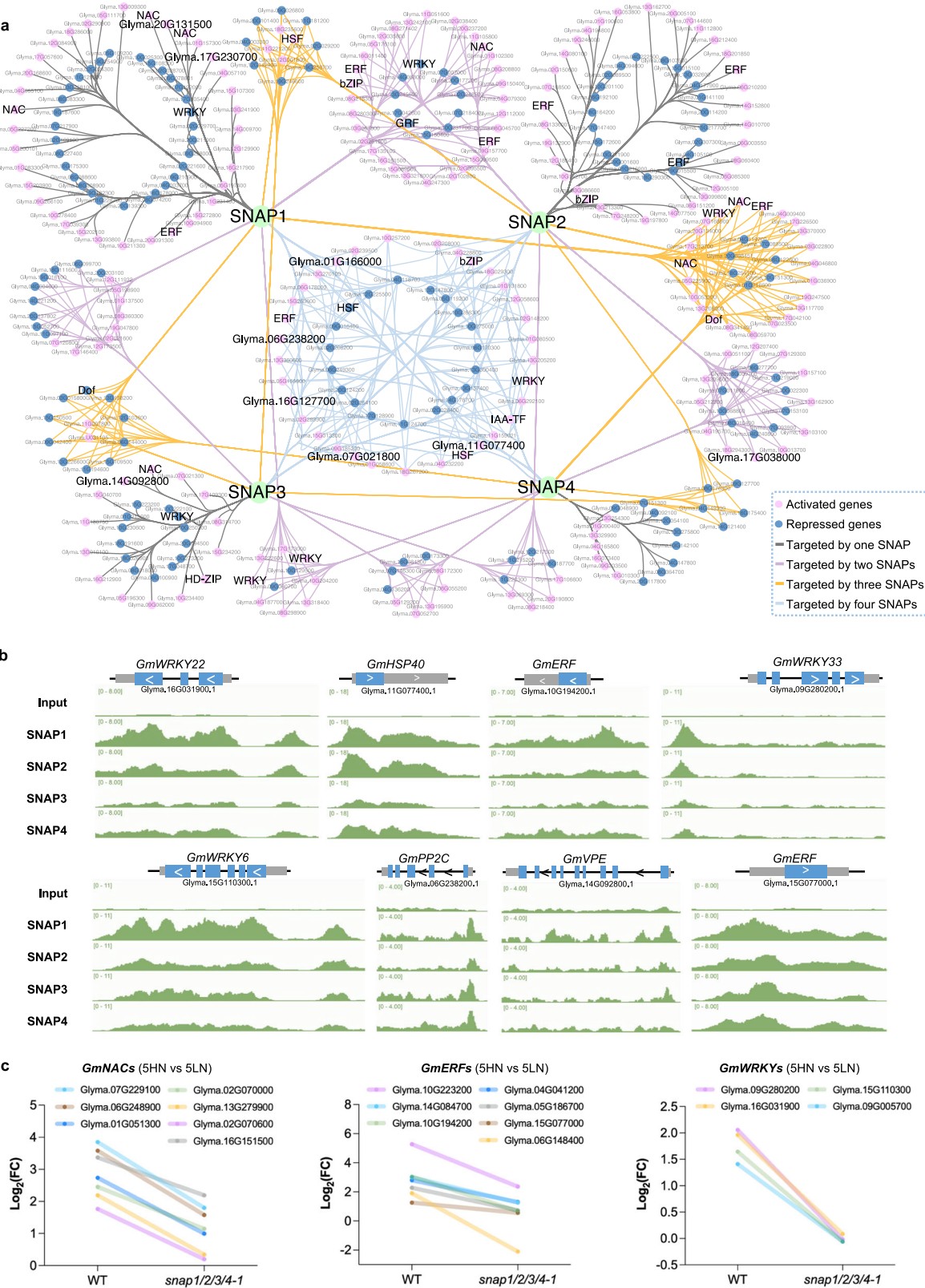

**Fig. 6 | SNAP TFs mediate a N-responsive transcription network. a** Network of N-responsive genes directly regulated by SNAP1/2/3/4. NAC: NAM, ATAF, and CUC; ERF ethylene responsive element binding factor, bZIP Basic-leucine zipper, IAA-TF IAA transcriptional regulator, HSF heat shock transcription factor, Dof DNA binding with One Finger, HD-ZIP homeodomain leucine zipper. **b** Examples of SNAP1/2/3/4 ChIP peaks in target genes. **c** Relative expression of TFs directly activated by SNAP1/2/3/4 in high vs low N treatments. FC fold change. Source data underlying **c** is provided as a Source Data file.

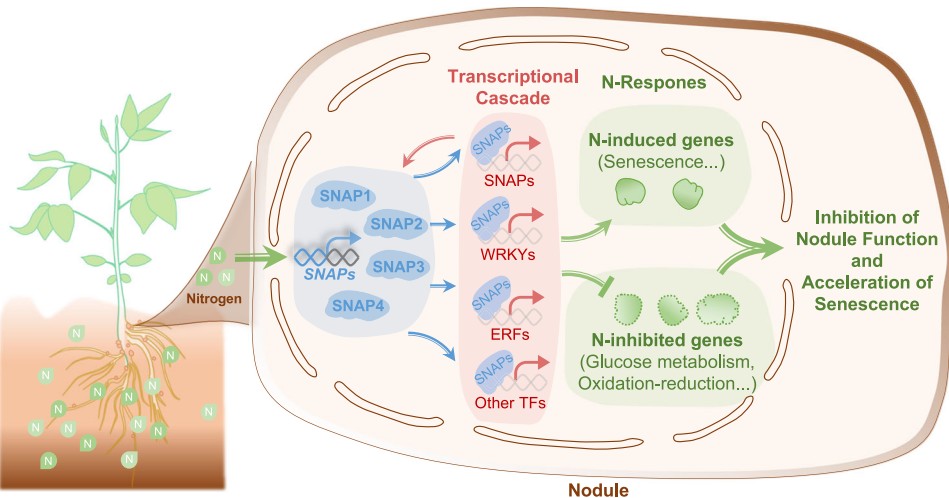

**Fig. 7 | Regulatory model of SNAP TFs in mature nodules under high N.** High N activates the expression of SNAP TFs which orchestrate a transcriptional cascade to promote early nodule senescence to repress N-fixation.

SNAP-mediated network would bring more new insights into N responses in the nodule.

Nodule senescence is a highly organized and genetically controlled process[18]. Besides high N, exposure to various environmental stresses, such as prolonged darkness, salt stress and drought stress, also induce nodule senescence[18], but in these cases little is known on how senescence is triggered[18]. It is possible that SNAPs may act as key hubs in response to various stresses and trigger the reprogramming of transcriptome toward the transition between N₂ fixation and nodule senescence. For instance, The SNAC-B/NAP subfamily NAP TF was recognized as a negative regulator of salt stress response in Arabidopsis[44]. GmNAC181, which was identified as SNAP4 in this study, may play a role in salt stress repression of soybean nodule number[45]. As the NAP clade TFs are evolutionarily conserved, it is possible that the NAP mediated nitrate signaling in the nodule may be conserved among legume species. In *Medicago truncatula*, MtNAC969 (Medtr4g081870), a homolog of SNAP10/11 (Supplementary Fig. 11), is also induced by nitrate in symbiotic nodule and is involved in nodule senescence[46,47].

It remains an open question how high N status is sensed and transduced to the SNAPs mediated gene regulatory network in the soybean nodule. In *M. truncatula* and *Lotus japonicus*, nitrate is taken up through NRT2 transporters to suppress nodulation[48,49]. The transporter NPF7.6 mediates transport of nitrate in transfer cells within nodule vascular bundle[49]. In *L. japonicus*, the uptake of nitrate leads to activation of LjNLP1, which enhances the expression of LjNRT2.1 which further increases nitrate uptake[48]. While NRT2s mediate nitrate uptake from the environment and are expressed in nodules[50], the role of these transporters in nodules remain unclear, and it remains to be shown whether nitrate acts directly or indirectly on N-fixing cells. Regardless, whether acting locally or systemically, it seems likely that the NRT-mediated and NLP-dependent signaling module acts upstream of the N- response in soybean nodules.

Leguminous plants evolved with negative regulatory systems to repress root nodule symbiosis in response to increased soil N availability[1]. When soil N is increased and fulfills the N demand of host plants, the nodule N-responsive transcriptional network could rapidly repress the plant-rhizobia mutual relationship to avoid unnecessary C investment into SNF. However, such mechanism also led to limitation of the efficacy of soybean nodulation in modern agricultural system, such as in China[5]. Chinese farmers thus rarely use commercial rhizobia to inoculate legumes like soybean (*Glycine max*) and rely instead on chemical N fertilization, resulting higher production costs and damage to the environment. The breeding of "N-insensitive SNF" soybean varieties might help optimize the current agricultural practice. A "N-insensitive SNF" soybean variety would ideally exhibit highly N tolerant nodulation, for nodule number, nodule growth, N₂ fixation, and nodule senescence. Several soybean AON mutants were identified that formed excessive nodules under high N, yet the nodules formed were small[9,10]. Here we show that the mature nodules of *snap1/2/3/4* quadruple mutants were relatively tolerant to high N exposure. Mutagenesis of SNAPs and AON genes together may alleviate N inhibition of both nodulation and SNF. Future studies should be performed on whether this strategy could facilitate the generation of "N-insensitive SNF" varieties with good agronomic traits, particularly in field conditions with heavy fertilization.

## Methods

### Plant material and growth conditions

Soybean (*Glycine max*) cultivar Williams 82 was used for physiological experiments and genetic transformation. Seeds were surface sterilized by the chlorine gas method and germinated on sterilized and soaked vermiculite. Five days after germination, seedlings were inoculated with the rhizobium strain *Bradyrhizobium* sp. BXYD3 and transferred to hydroponic culture. The medium composition was as follows: Ca(NO₃)₂ 0.12 mM, KNO₃ 0.19 mM, MgCl₂ 2.5 µM, MgSO₄ 0.5 mM, K₂SO₄ 1 mM, MnSO₄ 0.5 µM, ZnSO₄ 1.5 µM, CuSO₄ 0.5 µM, (NH₄)₆Mo₇O₂₄ 0.15 µM, KH₂PO₄ 0.25 mM, NaB₄O₇ 0.25 µM, Fe-EDTA 0.04 mM, (NH₄)₂SO₄ 0.05 mM, and CaCl₂ 1.2 mM. The pH was maintained at pH 5.8 with 1 M KOH every 2 d, and the medium was changed once a week. For high N medium, Ca(NO₃)₂ 1.2 mM and KNO₃ 1.9 mM were used. During high N treatments, high N or control medium was changed every 24 h. Soybean plants were grown in a growth chamber under the following conditions: light intensity of 450 µmol photons m⁻² s⁻¹, 14 h light at 28 °C and 10 h dark at 24 °C, humidity 65%. Soybean plants were treated with high N or control medium for 5 days after 25 days of germination and then transferred to soil for harvest.

### Transcriptomic analysis and quantitative real-time PCR

Total RNA was extracted from plant tissues using RNA-Solv® reagent (OMEGA Bio-Tek, Norcross, GA, United States). A total amount of 1 µg RNA per sample was used as input material for transcriptome deep sequencing and RT-qPCR. Sequencing libraries were generated using the NEBNext® Ultra™ RNA Library Prep Kit for Illumina® (NEB, USA) following the manufacturer's recommendations. The libraries were sequenced on an Illumina Novaseq platform. We used Trimmomatic v0.39 to assess the quality control of raw RNA-seq reads, trim adapter sequences[51]. The clean reads were aligned to the cultivated soybean Wm82 a4.v1 reference genome using HISAT2[52] v2.1.0. The number of

reads mapping to each gene and normalized expression value (FPKM) was calculated by StringTie[53] v1.3.6. Read count was used to perform differentially expression analysis using DESeq2[54] v1.34 with a false discovery rate (FDR) < 0.05 and | log$_2$(fold-change)| ≥ 1 between treatment and control groups. AgriGO was applied to perform Gene Ontology (GO) enrichment analysis[55].

For RT-qPCR, first-strand cDNA was synthesized using the PrimeScript RT reagent kit (TaKaRa, Japan), and PCR amplification was performed using SYBR1 Premix Ex Taq™ (TaKaRa). The soybean gene *translation elongation factor eEF-1alpha* (*TefS1*, accession number X56856) was used as reference to calculate the relative expression levels of each gene by the $2^{-\Delta\Delta CT}$ method[56].

## Co-expression network construction of transcription factors
Differentially expressed genes (DEGs) in at least one time point under high N treatment in nodule tissue were used to construct a network by using WGCNA R package v1.70.3[57]. Log$_2$(FPKM + 1) values of DEGs were used to calculate the adjacency matrix. The signed gene co-expression and ARA activity network was constructed by using soft threshold power of 9 and a minimum module size of 10. Modules with cut heights <0.3 were merged. Intramodular connectivity (sum of the weight of intramodular edges) and gene trait significance (correlation between gene expression pattern and ARA activity change) of each transcription factor (TF) were calculated to rank them in 'blue' module. The overlapped TFs with top 20 intramodular connectivity and gene trait significance were selected as hub TFs. Subnetworks of selected modules were visualized in Cytoscape v3.5[58] by filtering weighted correlation value of 0.27 with TF node for better visualization.

## Chromatin immunoprecipitation sequencing (ChIP-seq)
Soybean *SNAP* transgenic hairy root tissues with nodules were fixed with 1% formaldehyde in PBS for 15 min under vacuum. After nuclei extraction, the chromatin was sonicated to 300-500 bp by using Covaris M220. The chromatin samples were incubated with Dynabeads protein A/G (Invitrogen) with FLAG antibody (Sigma, F1804) for 6 h. The beads were then washed twice with low salt buffer (150 mM NaCl in TE) and twice with high salt buffer (250 mM NaCl in TE). The washed beads were treated with 1 uL of Tn5 transposase in 1x Tn5 buffer (10 mM Tris-HCl pH 8.0, 5 mM MgCl$_2$) at 37 °C for 30 min. The beads were then washed with TE buffer and reverse-crosslinked for 8 h. The purified DNA was then amplified using N50x and N70x index primers. Two biological replicates were performed for each experiment.

Paired-end raw reads were trimmed by trim_galore[59] v0.6.6 for adapters and low quality with default parameters and high-quality reads were then aligned to the cultivated soybean Wm82 a4.v1 reference genome using Bowtie2[60] v2.3.4.3 with very-sensitive model. Properly paired and uniquely mapped reads were then filtered by mapping quality (MAPQ > 30) using samtools[61] v1.9 for subsequent analyses. Read coverage was calculated using bamCoverage function in deeptools[62] v3.5.1 with bin size of 1. Peak calling was performed by MACS2[63] v2.2.7.1 with parameters of -q 0.05 and -g 1.0e9. HOMER[64] v4.11.1 was applied to predict the enriched motifs for candidate peaks with default parameters. Finally, peak annotation was conducted by annotatePeaks.pl script in Homer and target genes were selected by harboring peaks within 2 kb up- or downstream from the transcription start site of gene model.

## Generation of vectors and transgenic plants
For CRISPR/Cas9 mutagenesis, three sgRNA were designed targeting the *SNAP* coding region. Vector construction were performed as described with pGES401[65]. Then, the plasmids carrying the sgRNA cassettes were pooled transformed into Wm82 via the cotyledonary-node transformation method as described[66]. Two quadruple *SNAP* mutants were isolated from two independent transgenic lines and validated by Sanger sequencing. *snap1234-1* carried 65 bp deletion,

536 bp deletion, 54 deletion and 587 bp mutation consisting of 585 inversion, 1 bp deletion and 1 bp insertion respectively. *snap1234-2* carried 13 bp deletion, 2 bp deletion, 43 bp deletion and 1 bp deletion respectively.

To construct the expression vector for ChIP-Seq analysis, the CDS fragment of each *SNAP* was cloned from cDNA of Williams 82 and fused with the FLAG-tag into the expression vector pTransE1 drived by the soybean pM4 promoter using Gateway method. Then, four the expression vectors were transformed into *Agrobacterium tumefaciens* strain K599 for soybean hairy root transformation using hypocotyl transformation. After incubation about 2 weeks, the positive hairy root was identified using hand-held laser gun and collected into 50-mL centrifuge tube.

## Physiological observations
For SYTO9 observation, nodules from soybean root were collected and embedded within 5% agarose. After solidification, samples were sliced using a Leica RM2255 microtome in directions of 70-μm thickness. The nodule median sections were selected and further stained with SYTO9 (Life, s34854) staining solution for 7–10 min in dark condition. Then, nodule images were taken by Leica SP8 confocal microscope for SYTO9 stain observation. All images in a single experiment were obtained with the same settings. The fluorescence intensity of each image was calculated by imageJ software[67].

For nodule color observation, soybean root nodules were selected and cut open in the middle with a blade. Then, nodule samples were immediately observed using stereoscopic microscope. All images in a single experiment were obtained with the same settings.

Nitrogenase activity was determined by the acetylene reduction assay (ARA) by gas chromatography (GC-14, Japan). Nodule samples with similar size were collected in 12 mL sealed glass bottles. One milliliter gas was extracted from the bottle with a syringe and then injected 1 ml acetylene. The sealed bottle was put in room temperature for 2 h. 1 mL mol·L$^{-1}$ NaOH was added to the bottle for terminating the reaction with a syringe and then the bottle was sealed. Finally, 1 mL gas was extracted from the bottle to detect the amount of ethylene by gas chromatography.

The measurement of leghemoglobin content of nodule was adopted from previous study[19]. About 0.2 g of fresh nodule tissue was grounded and homogenized in 0.1 mol·L$^{-1}$ phosphate buffer (PBS, pH 6.8 at 4 °C). After centrifuged for 15 min at 100 × g, the supernatant was transferred and centrifuged again at 12,000 × g for 20 min. Absorbance of the supernatant were measured at 520, 540, and 560 nm using PBS as blank control.

Nodule samples for TEM observation were prepared as followed. Nodule samples were cut into ~2 mm long and fixed in 0.1 M phosphate buffer (pH = 7.2) containing 3% glutaraldehyde and 2.5% paraformaldehyde. The samples were kept in 4 °C after vacuuming 15–30 min. After six times washing with 0.1 M phosphate buffer, the nodule samples were post-fixed in 1% osmium tetroxide for 4 h and washed with 0.1 M phosphate buffer. Next, fixed nodule samples were dehydrated and embedded in flat molds using EPON812 resin. Ultrathin sections (70 nm) were cut by ultramicrotome (Leica UC7). Ultrathin sections were observed by a transmission electron microscope (FEI Tecnai G2 Spirit BioTWIN) operating at 100 KV.

## Statistics and reproducibility
In this study, statistical analysis was performed using GraphPad Prism 9 (v9.0.0). No data were excluded from the analyses and no statistical method was used to predetermine sample size. For microscopic and physiological observations, as least two completely independent experiments were performed. At least three biological replicates were measured for each experiment. For minimize plant-to-plant variations, at least 7 individual plants were used in each experiment for sample collection. For RNA-seq data and ChIP-seq data, three biological

experiments and two biological experiments were performed, respectively. Raw data underlying the figures are provided as a Source Data file.

### Reporting summary

Further information on research design is available in the Nature Portfolio Reporting Summary linked to this article.

## Data availability

Data deposition: The raw sequence data generated in this study have been deposited in the National Center for Biotechnology Information (NCBI) under the BioProject accession number PRJNA674706 and PRJNA907509. Processed data have been deposited in the NCBI GEO database under the accession number GSE236557. The Wm82 a4.v1 reference genome was download from Phytozome. Source data are provided with this paper.

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

## Acknowledgements

We thank the Microscopy Core Facility at the Horticultural Plant Biology and Metabolomics Center (HBMC). We thank Rufang Deng from South China Botanical Garden, Chinese Academy of Sciences for helps with TEM sample preparation and observation. We thank Professor Hong Liao for helps with ARA assay. This work was supported by the National Key Research and Development Program (grant no. 2022YFA0912100) to X.Z. and the National Natural Science Foundation of China (32072083, 31772379) to Y.G. The authors declare no conflict of interest.

## Author contributions

Y.G. and P.L. coordinated and designed the project. X.W., Z.Q., WZ., N.W., and C.C. performed experiments and data analysis. H.K., M.B., and Z.Q finished vector constructions and transformation. X.B.Z. and F.K. helped interpret the results. Y.G., P.T.L., X.W., Z.Q., W.Z, and N.W. wrote the manuscript.

## Competing interests

The authors declare no competing interests.
