## [Peer Review File · Nature Communications]

The NAC transcription factors SNAP1/2/3/4 are central regulators mediating high nitrogen responses in mature nodules of soybeanREVIEWER COMMENTS

Reviewer #1 (Remarks to the Author):

The manuscript describes an thorough analysis of N-induced nodule senescence in soybean nodules, providing insights into the molecular mechanisms that control this process. The authors define a set of transcription factors that at least in part they propose are central to the activation of senescence following the perception of external N. The work is thorough, the paper well written and the discoveries novel.

My concerns are listed below:

1. The authors need to be considerably more careful in the use of the term significant. They make statements about results being significant, when such results are only qualitative, not quantitative: lines 116, 118, 213. In these cases no statistics is performed on the data being described
2. Following on from concern 1: I think the authors need to make some of their measures more quantitative, especially measures of nodule senescence. This is central to the conclusions of the paper, but is currently being based on a slight green coloration to WT nodules in high N and to single microscope images of relatively low-resolution micrographs of nodules. The colour of nodules can and should be quantified and a measure defined for the prevalence of functioning symbiosomes. Currently single images are shown: much more images could be added to the supplementary data and if possible quantification brought to bear. TEMs of nodule sections would be much more informative than the current low resolution images using syto9 staining.
3. Lines 148-151. These are bold conclusions based on GO enrichment and in my opinion the authors are overstating what is possible to infer from these GO-enrichments.
4. Line 214-216. This is a massive overstatement of the results. The N response is not alleviated in the mutant nodules, it is reduced, but not alleviated.
5. Line 193. It appears a claim is being made here that is not statistically significant in the data presented. The nodule number should only be claimed lower if statistically significant

6. Lines 217-219. I don't understand the comparison being made here. It seems inconsistent with their conclusions, but perhaps I misunderstand the exact comparison. Please clarify the comparison being made

7. Lines 316-318. The authors are claiming a novel mechanism for N-regulation in nodules. However, they have not tested the snap mutants in any other context or tissue except nodules. To support this claim the authors should test N-response in the snap mutants in other tissues.

8. lines 336-337. This statement needs citations

9. In the abstract and discussion the authors suggest that N-insensitive nodulation could lead to higher yields. I doubt this very much. The authors need to higher remove this claim or better justify it.

10. Overall the paper is very well written, however, the abstract needs some attention in wording and grammar.

Reviewer #3 (Remarks to the Author):

The authors investigated the repressive effects of high nitrogen on nodulation and symbiotic nitrogen fixation (SNF). First, they employed a hydroponic culture system for reversible N treatment. The reversible inhibition of SNF has been tested via (1) acetylene reduction assay (ARA); (2) nodule colorization; and (3) rizobia viability. Second, they performed bulk RNA-Seq analysis on the samples taken 1, 3, and 5 days after high N treatment and/or 2 days after low N. It is an excellent example of top-down analysis of a large group of differentially expressed genes that resulted in meaningful insight into the underlying network guided by a cluster of NAC TFs, here named SNAP. They generated quadruple mutants, which were indistinguishable from the wild type at low N conditions but displayed less sensitivity to high N. They also profiled the transcriptome of the mutant and performed a TF binding assay for key SNAPS, describing quite a complex network of SNAP-mediated signaling.

In general, it is a nicely designed and performed study, well written and visualized. The work is of significance to the plant biology and agricultural sciences.

My main concern after reading is a discrepancy between the vast SNAP-mediated network that has been detected (Fig. 5A) and the proposed model where the role of SNAPs is limited to inhibition of nodule formation upon response to high N. The question is why have plants evolved such a remarkable mechanism with great functional redundancy for a condition that did not happen before the heavy application of N fertilizers in agricultural practices?

Although, the authors suggest using the newly discovered regulators for the creation of "N-insensitive SNF", at the moment I doubt the conclusion that the quadruple mutant is indistinguishable from the wild type at low N. I suggest a bit more detail be given on the mutant phenotyping to describe N-unrelated effects.

Minor thing:

The overlap between SNAP-binding genes and N-responsive genes does not look profound (lines 243-249; Fig. 4h). How does this data look when compared with the RNA-Seq datasets made on the wild type? Is there any enrichment for SNAP targets at any time point/condition?

There are a few style and punctuation mistakes over the text, e.g., line 99 "samples were sampled", line 148 with two dots, etc.

Also, lines 139 and 141: consider to put the module names in " ('turquoise' module).

Response:

We appreciate the referees for the constructive comments on our manuscript. In this revised manuscript, we have made substantial revisions in response to the comments, highlighted yellow in the text. Below please find the point-by-point response, which we hope could answer the concerns.

Reviewer #1 (Remarks to the Author):

The manuscript describes an thorough analysis of N-induced nodule senescence in soybean nodules, providing insights into the molecular mechanisms that control this process. The authors define a set of transcription factors that at least in part they propose are central to the activation of senescence following the perception of external N. The work is thorough, the paper well written and the discoveries novel.

My concerns are listed below:

1. The authors need to be considerably more careful in the use of the term significant. They make statements about results being significant, when such results are only qualitative, not quantitative: lines 116, 118, 213. In these cases no statistics is performed on the data being described

Response: Thank you for the helpful suggestion. In the revised manuscript, we measured the content of red leghemoglobin to quantify the nodule color parameter (lines 113-114, Figure 1d and Figure S2a). Also, the SYTO9 fluorescence intensity was statistically analyzed for live bacteroids in infected cells (lines 114-116, Figure 1f and Figure S2b). These statistical analyses were consistent with the visual observations that high N accelerated the senescence of WT nodules, and *snap 1/2/3/4* mutants were less sensitive to high N (lines 201-205, Figure 4b and 4d). For line 213 in the previous manuscript, we have added statistical test results to Figure 5b (lines 220-222).

2. Following on from concern 1: I think the authors need to make some of their measures more quantitative, especially measures of nodule senescence. This is central to the conclusions of the paper, but is currently being based on a slight green coloration to WT nodules in high N and to single microscope images of relatively low-resolution micrographs of nodules. The colour of nodules can and should be quantified and a measure defined for the prevalence of functioning symbiosomes. Currently single images are shown: much more images could be added to the supplementary data and if possible quantification brought to bear. TEMs of nodule sections would be much more informative than the current low resolution images using syto9 staining.

Response: Thank you for the suggestions. We have quantified the leghemoglobin content and fluorescence intensity of nodule samples under different N treatments to infer the change of nodule color, nitrogen fixation activity and live bacteroids (Figure 1d and 1f, Figure 4b and 4d and Figure S2, lines 113-116, lines 201-205). More raw images have been added to Figure S3. Moreover, we performed TEM and observed

typical senescence symptoms of high N exposed nodules and the reduced sensitivity of *snap* mutants to high N. These results were added to Figure 1g, Figure 4e, Figure S4, and Figure S5 (lines 117-126, lines 205-208). We have revised the content accordingly.

3. Lines 148-151. These are bold conclusions based on GO enrichment and in my opinion the authors are overstating what is possible to infer from these GO-enrichments.

Response: We have removed this interpretation to avoid overstatement from the GO enrichment of module turquoise (lines 151-155).

4. Line 214-216. This is a massive overstatement of the results. The N response is not alleviated in the mutant nodules, it is reduced, but not alleviated.

Response: We have rewritten the sentence to “These results indicate that the transcriptional responses to high N are reduced in *snap1/2/3/4* nodules.” (lines 222-223).

5. Line 193. It appears a claim is being made here that is not statistically significant in the data presented. The nodule number should only be claimed lower if statistically significant

Response: Thank you for the suggestion. We have rewritten the sentence to “The mutant nodule number and the average nodule weight was not statistically different from the WT.” (lines 196-198).

6. Lines 217-219. I don't understand the comparison being made here. It seems inconsistent with their conclusions, but perhaps I misunderstand the exact comparison. Please clarify the comparison being made

Response: In this paragraph, we showed that more DEGs of “*snap* VS WT” were found at 5HN (3,282) than that at low N condition (1,096), indicating that SNAP1/2/3/4 tend to affect nodule transcriptome after high N exposure. Together with the finding that 69% of the 5HN “*snap* VS WT” DEGs were identified as N-responsive genes in WT nodules, we concluded that SNAP1/2/3/4 TFs mainly influence the N-responsive transcriptome in mature nodules.

We have rewritten the sentence to “We next compared the transcriptome between *snap1/2/3/4* mutant and WT nodules at low N and high N conditions, respectively. Compared to 1,096 DEGs at low N condition, we found more DEGs (3,282) between *snap1/2/3/4* and WT in 5HN nodules (Figure 5c and Table S9 and S10). This result indicated that SNAP1/2/3/4 makes more impact on nodule transcriptome after high N exposure.” (lines 224-228).

7. Lines 316-318. The authors are claiming a novel mechanism for N-regulation in nodules. However, they have not tested the *snap* mutants in any other context or tissue

except nodules. To support this claim the authors should test N-response in the snap mutants in other tissues.

Response: Thank you for the suggestion. We agree that our results may not be sufficient to propose a novel N-response mechanism in tissues other than nodule. Considering that elucidating the role of SNAPs in other tissues may be out of the focus of this manuscript, we have rewritten the sentence to avoid overstatement.

Revised sentence: "In this work, we found that the high N response of mature nodules may be mediated by a SNAPs-centered transcriptional network which activates WRKY, ERF and NAC family TFs." (lines 323-325).

8. lines 336-337. This statement needs citations

Response: The citations were added to the text. (line 352).

9. In the abstract and discussion the authors suggest that N-insensitive nodulation could lead to higher yields. I doubt this very much. The authors need to higher remove this claim or better justify it.

Response: Thank you for the suggestion. We have rewritten related sentences and removed the perspective about increasing yield.

Line 45-47 "Thus, breeding soybean varieties with N₂ fixation ability that is stable across a range of N levels may provide environmental benefits through decreasing N-inputs."

Line 368-369 "We propose that the creation of "N-insensitive SNF" soybean varieties using molecular approaches might help optimize the current agricultural practice."

10. Overall the paper is very well written, however, the abstract needs some attention in wording and grammar.

Response: Thank you for the suggestion. We have revised the abstract.

Reviewer #3 (Remarks to the Author):

The authors investigated the repressive effects of high nitrogen on nodulation and symbiotic nitrogen fixation (SNF). First, they employed a hydroponic culture system for reversible N treatment. The reversible inhibition of SNF has been tested via (1) acetylene reduction assay (ARA); (2) nodule colorization; and (3) rizobia viability. Second, they performed bulk RNA-Seq analysis on the samples taken 1, 3, and 5 days after high N treatment and/or 2 days after low N. It is an excellent example of top-down analysis of a large group of differentially expressed genes that resulted in meaningful insight into the underlying network guided by a cluster of NAC TFs, here named SNAP. They generated quadrapule mutants, which were indistinguishable from the wild type at low N conditions but displayed less sensitivity to high N. They also profiled the

transcriptome of the mutant and performed a TF binding assay for key SNAPs, describing quite a complex network of SNAP-mediated signaling.

In general, it is a nicely designed and performed study, well written and visualized. The work is of significance to the plant biology and agricultural sciences.

My main concern after reading is a discrepancy between the vast SNAP-mediated network that has been detected (Fig. 5A) and the proposed model where the role of SNAPs is limited to inhibition of nodule formation upon response to high N. The question is why have plants evolved such a remarkable mechanism with great functional redundancy for a condition that did not happen before the heavy application of N fertilizers in agricultural practices?

Response: Thanks for this comment. In root nodule symbiosis, leguminous plants need to provide substantial C sources to the nodule in exchange for fixed N. For example, nodules consume 9% of the net photosynthates produced by cowpea plants. Therefore, leguminous plants evolved with negative regulatory systems to repress root nodule symbiosis in response to the fluctuating N availability in the natural system (Streeter and Wong 1988; Nishida and Suzaki 2018). When soil N is sufficient to fulfill the N demand of host plants, the nodule N-responsive transcriptional network could repress the plant-rhizobia mutual relationship to avoid unnecessary C investment into nodules. However, such a mechanism also led to the inhibition of root nodule symbiosis by heavy N fertilization in modern agricultural systems. In the revised manuscript, we added additional discussions regarding this concern. (Lines 360-365).

Streeter, J., & Wong, P. P. (1988). Inhibition of legume nodule formation and N₂ fixation by nitrate. *Critical Reviews in Plant Sciences*, 7(1), 1-23.

Nishida, H. & Suzaki, T. Two Negative Regulatory Systems of Root Nodule Symbiosis: How Are Symbiotic Benefits and Costs Balanced? *Plant Cell Physiol* 59, 1733-1738 (2018).

Although, the authors suggest using the newly discovered regulators for the creation of "N-insensitive SNF", at the moment I doubt the conclusion that the quadruple mutant is indistinguishable from the wild type at low N. I suggest a bit more detail be given on the mutant phenotyping to describe N-unrelated effects.

Response: Thank you for the suggestion. We have added more phenotypic observations to Figure S12, including plant height, flowering time, branch number and node number of plants. No significant differences were observed between *snap1/2/3/4* quadruple mutants and WT at low N condition at the vegetative stage. Nevertheless, we still could not rule out the possibility of other unexamined traits being affected in *snap1/2/3/4* mutants. We would keep with awareness of the phenotype of *snap1/2/3/4* mutants in the future.

Minor thing:

The overlap between SNAP-binding genes and N-responsive genes does not look profound (lines 243-249; Fig. 4h). How does this data look when compared with the RNA-Seq datasets made on the wild type? Is there any enrichment for SNAP targets at any time point/condition?

Response: Thank you for the comment. In the presented overlap between SNAP-binding genes and snap1/2/3/4 influenced N-responsive genes, we identified 17.0% (388 of 2279) N-responsive genes to be directly regulated by SNAP1/2/3/4. In the revised manuscript we also integrated SNAP-binding genes with WT N-responsive genes at 5HN, and found 13.3% (1254/9422) DEGs were SNAP binding targets. Therefore, the SNAP binding targets tend to be enriched in snap1/2/3/4 influenced N-responsive genes, in comparison with total N-responsive genes in WT nodules. This is consistent with the role of SNAPs in regulating the expression of downstream targets.

It is interesting that SNAP1/2/3/4 directly binds to a small portion of SNAPs influenced N-responsive genes. Therefore, SNAP1/2/3/4 may largely influence N-responsive genes by orchestrating a TF hierarchy in the nodules. Alternatively, we could not rule out the possibility of transient bindings which may affect target gene expression yet elude biochemical detection. Relative discussions were added to the revised manuscript. (Lines 323-330).

There are a few style and punctuation mistakes over the text, e.g., line 99 "samples were sampled", line 148 with two dots, etc.

Also, lines 139 and 141: consider to put the module names in " ('turquoise' module).

Response: Thank you for the suggestion. We have corrected the mistakes and checked the content more carefully.

REVIEWERS' COMMENTS

Reviewer #1 (Remarks to the Author):

The authors have done an excellent job addressing my prior concerns. This is an important paper, well written and thorough.

Giles Oldroyd

Reviewer #2 (Remarks to the Author):

The authors elaborated on the questions raised at the first step of the reviewing process. They provided detailed explanations to support their arguments. I have no further questions.